# Successful Treatment of Persistent SARS-CoV-2 Infection in a B-Cell Depleted Patient with Activated Cytotoxic T and NK Cells: A Case Report

**DOI:** 10.3390/ijms222010934

**Published:** 2021-10-10

**Authors:** Jacek Jassem, Natalia Maria Marek-Trzonkowska, Tomasz Smiatacz, Łukasz Arcimowicz, Ines Papak, Ewa Jassem, Jan Maciej Zaucha

**Affiliations:** 1Department of Oncology and Radiotherapy, Medical University of Gdańsk, 80-214 Gdańsk, Poland; 2International Centre for Cancer Vaccine Science, University of Gdańsk, 80-822 Gdańsk, Poland; natalia.marek-trzonkowska@ug.edu.pl (N.M.M.-T.); lukasz.arcimowicz@phdstud.ug.edu.pl (Ł.A.); ines.papak@phdstud.ug.edu.pl (I.P.); 3Laboratory of Immunoregulation and Cellular Therapies, Department of Family Medicine, Medical University of Gdańsk, 80-210 Gdańsk, Poland; 4Department of Infectious Diseases, Medical University of Gdańsk, 80-214 Gdańsk, Poland; tomasz.smiatacz@gumed.edu.pl; 5Department of Pneumonology, Medical University of Gdańsk, 80-214 Gdańsk, Poland; ejassem@gumed.edu.pl; 6Department of Haematology, Medical University of Gdańsk; 80-214 Gdańsk, Poland; jan.zaucha@gmail.com

**Keywords:** COVID-19, oligoclonal T cell response, TCR Vβ repertoire, CD8+ cytotoxic T cells, natural killer cells (NK cells)

## Abstract

We report a lymphoma patient with profound B-cell deficiency after chemotherapy combined with anti-CD20 antibody successfully treated with remdesivir and convalescent plasma for prolonged SARS-CoV-2 infection. Viral clearance was likely attributed to the robust expansion and activation of TCR Vβ2 CD8+ cytotoxic T cells and CD16 + CD56- NK cells. This is the first presentation of TCR-specific T cell oligoclonal response in COVID-19. Our study suggests that B-cell depleted patients may effectively respond to anti-SARS-CoV-2 treatment when NK and antigen-specific Tc cell response is induced.

## 1. Introduction

Immunocompromised patients carry a particularly high risk of COVID-19-related mortality [1]. These patients may also show prolonged SARS-CoV-2 shedding, but its late clinical consequences are unclear [2]. The efficacy of currently available antiviral therapies and the optimal management of immunocompromised patients remains to be established. We report a case of a B-cell deficient lymphoma patient with protracted shedding of viable SARS-CoV-2, successfully treated with the combination of remdesivir and convalescent plasma (CP) three months after primary infection. Detailed immunologic analysis showed a unique TCR-specific oligoclonal response of T cells. We also discuss the role of natural killer (NK) and T cell activation in resolving SARS-CoV-2 infection in immunocompromised patients.

In 2018, a 66-year-old Caucasian male was diagnosed with stage IV transformed follicular lymphoma. The patient had no other comorbidities, did not smoke, had a BMI of 25 kg/m^2^, and was in a good general condition and professionally active. He received six cycles of cyclophosphamide, doxorubicin, vincristine, prednisone, and rituximab, intercalated with two cycles of high-dose methotrexate CNS prophylaxis. A complete metabolic response was achieved, and the patient was subjected to maintenance treatment with the CD20^+^ cell-depleting antibody obinutuzumab administered every 2–3 months. Treatment resulted in immunosuppression manifested by moderate hypogammaglobulinemia (IgG1 5.20 G/L, IgG2 0.99 G/L, IgG3 0.13 G/L, and IgG4 0.11 G/L), and recurrent respiratory infections. Although total lymphocyte counts were not decreased (1.02 × 10^9^/L; normal range [NR], 0.9–5.0 × 10^9^/L), the CD19^+^ B cells were undetectable. The CD3^+^ T cells comprised 90% of the lymphocytes (0.92 × 10^9^/L; NR, 0.61–2.25 × 10^9^/L). The count of CD4^+^ helper T (Th) cells was decreased (0.17 × 10^9^/L, NR, 0.43–1.51 × 10^9^/L) and CD8^+^ cytotoxic T (Tc) cells were within NR (0.68 × 10^9^/L, NR, 0.1–0.84 × 10^9^/L). Thus, Th cells accounted for 17% (NR, 32–67%) and Tc cells for 67% of lymphocytes (NR, 8–40%). NK cell counts were 0.09 × 10^9^/L (NR, 0.11–0.65 × 10^9^/L) and constituted 9% of lymphocytes (NR, 6–35%).

On September 25, 2020 (day 1), the patient presented with a dry cough, fever, and malaise, and on day 2, a nasopharyngeal/throat swab tested positive for SARS-CoV-2 by reverse-transcription polymerase chain reaction (qRT-PCR; Figure 1). All symptoms spontaneously subsided within a few days, but consecutive nasopharyngeal swabs on days 15, 24, and 35 remained positive. On day 40, the nasopharyngeal antigen assay (Roche Diagnostics, Indianapolis, IN, USA) tested negative. Hence, the persistent positive qRT-PCR test was attributed to the shedding of residual, not viable, viral RNA. The patient remained asymptomatic and stayed with his family without transmission precautions. On day 53, the seventh dose of obinutuzumab was administered.

On day 72, mild intermittent malaise and chills occurred, followed by a fever up to 38.0 °C. C-reactive protein was 25 mg/L, D-dimer level was 0.502 µg/mL fibrinogen equivalent units, lymphocyte count dropped to 0.83 × 10^9^/L, oxygen saturation ranged from 95% to 98%, but the physical examination was unremarkable. On day 79, a high-resolution computed tomography (HRCT) of the chest showed left-side lobular pneumonia with ground-glass opacities and slight pleural effusion (CORADS 3; Figure 2C). The Roche SARS-CoV-2 antigen test was negative. Empirically administered levofloxacin and intravenous polyclonal immunoglobulins (30 g) proved clinically ineffective. On day 84, qRT-PCR from bronchoalveolar lavage tested positive for SARS-CoV-2, and comprehensive microbiological tests (cultures for aerobic and anaerobic bacteria, fungi and tuberculosis, and PCR multiplex for 22 viral and 33 bacterial pathogens) were negative. 

On days 87–96, the patient received 10 days of remdesivir (200 mg on day 1, followed by 100 mg daily), two 200 mL units of convalescent plasma (CP; day 88), and low-molecular-weight heparin, with good tolerance. All symptoms subsided rapidly, and an HRCT of the chest on day 118 showed complete resolution of inflammatory changes (Figure 2D). Two consecutive qRT-PCR tests from nasopharyngeal swabs on day 96, and four subsequent tests on days 106–187, proved negative. However, serological testing (Abbott) on days 96 and 116 did not detect IgG and IgM antibodies against SARS-CoV-2.

Three weeks after the negative qRT-PCR test, the patient underwent profound immune profiling. Study of repertoire of TCR Vβ clones was performed with flow cytometry for Th, Tc, and regulatory T cells (Tregs) in combination with detailed analysis of their activation markers. In addition, natural killer (NK) cells were divided into five subsets and tested for expression of cytotoxic mediators. Simultaneously, peripheral blood mononuclear cells (PBMC) from three healthy donors (HD) matched for age and gender without a history of COVID-19 were analyzed for comparison. 

On days 129 and 150, the patient received SARS-CoV-2 vaccination with Pfizer vaccine, however, IgG seroconversion has not been achieved. Three months after the recurrence of clinical infection, the patient has remained asymptomatic. 

## 2. Results

### 2.1. Distribution of Lymphocyte Subsets

After resolution, the patient differed strikingly from the HD in proportions of B cells, CD4+ Th cells, and CD8+ Tc cells. The frequency of particular lymphocyte subsets in the peripheral blood of the patient and HD were as follows: 0.3 vs. 11.0% (B cells), 20.5 vs. 38.6% (Th cells), 62.5 vs. 30.4% (Tc cells), 2.3 vs. 2.1% (Tregs), and 8.0 vs. 15.9% (NK cells), respectively (Figure 2A). The ratio of Th: Tc cells was 0.32 (NR 0.6–3.6). All tested T cell subsets of the patient were characterized by significantly lower numbers of TN cells, with the domination of TEMRA cells in the Tc cell population, compared with HD (67.2% vs. 34.2 %, respectively; Figure 2B). 

### 2.2. Expansion and Activation of TCR Vβ2 Clones of Tc Cells

Analysis of the TCR repertoire revealed significant oligoclonal expansion within Tc and Th cell populations of the patient, but not in the HD (Figure 3A, Appendix A). TCR Vβ22 and Vβ17 families accounted for 11.6% and 11.3% of the patient’s Th cells, respectively, compared to 3.8% and 6.1%, respectively, in the HD (Figure 3A, Appendix A). Within Tc cell repertoire, the most intensive expansion was found in clones expressing Vβ2 and Vβ23 TCR (16.7% and 7.4%, respectively), compared to the HD (3.7% and 0.9%, respectively; Figure 3A, Appendix A). The patient’s TCR Vβ2 positive Tc cells expressed markers of strong and prolonged activation. TN and TEMRA cells in the patient constituted 4.1% and 83.7% of TCR Vβ2 positive Tc cells, respectively, compared to 40.1% and 16.6%, respectively, in the HD (Figure 3B). The frequency of Vβ2 TCR positive Tc cells with simultaneous expression of perforin and 4–1BB, a costimulatory receptor induced by a TCR-specific signal [3], was four-fold higher in the patient than in the HD (82.1% vs. 20.7%, respectively; Figure 3B and Appendix A). In addition, 40.8% and 24.8% of TCR Vβ2 positive Tc cells were also positive for both IL-10 and IFN-β in the patient and HD, respectively (Figure 3B and Appendix A).

### 2.3. Alterations in the Distribution of NK Cells and Their Activation Status

Proportions of the following five NK cell subsets [4] were analyzed: CD3-CD56dimCD16+, CD3-CD56brightCD16-, CD3-CD56dimCD16-, CD3-CD56brightCD16dim, and CD3-CD56-CD16+. The main alterations were found in the frequency of the CD3-CD56-CD16+ and CD3-CD56dimCD16+ NK cell subsets that accounted for 21.7% and 73.0% of all NK cells, respectively, in the patient and 6.0% and 87.1%, respectively, in the HD (Figure 3C and Appendix A). A total of 83.4% and 44.7% of CD3-CD56-CD16+ cells expressed granzyme A and were negative for perforin in the patient and in HD, respectively. However, the numbers of CD3-CD56-CD16+ NK cells ne-gative for both perforin and granzyme A accounted for 14.5% in HD and only 1.0% in the patient (Figure 3C and Appendix A).

## 3. Discussion

Treatment options for COVID-19 are essentially limited to intravenous remdesivir and CP, and the efficacy of both approaches is still debatable [5,6,7]. Remdesivir aims to reduce viral load before the virus triggers a potentially devastating hyperinflammatory response. In immunocompetent patients, this compound is routinely administered within the first 7–10 days after symptom onset, as afterward viral replication wanes spontaneously, and later therapy does not prevent potential cytokine storm [7]. 

Patients with hematologic malignancies show complex and increasing immunologic consequences of SARS-CoV-2, including T cell exhaustion, the vulnerability of malignant B cells, and impaired seroconversion [8]. Experience with SARS-CoV-2 treatment in this group is scarce (Table 1).

In three patients with chronic lymphocytic leukemia (CLL) and febrile COVID-19 infection, viral clearance and fever resolution were achieved after treatment with CP [2,9,10]. In two of these patients, previous therapy with remdesivir provided only a short-term clinical improvement [9,10]. The transient effect of remdesivir therapy and viral clearance with CP was also reported in two follicular lymphoma patients [11,12]. Finally, viral clearance with CP after two unsuccessful attempts of remdesivir therapy was also achieved in a patient with primary immunodeficiency related to X-linked agammaglobulinemia [13]. The only individual reported who remained SARS-CoV-2 positive after combined treatment with remdesivir and CP was a patient on palliative chemotherapy diagnosed with advanced mantle-cell lymphoma [14]. All presented patients experienced persistent fever, pneumonitis, and prolonged viral shedding for more than 40 days, but none progressed to acute respiratory distress syndrome or multi-organ involvement. Despite long-term infection, only one of eight patients developed detectable anti–SARS-CoV-2 IgG levels.

Nevertheless, most of these reports focused on adaptive humoral immunity and included analysis of anti-SARS-CoV-2 antibodies and B-cell deficiencies and did not describe the entire immune status of the patients. In contrary, we focused on two main effector cell types known to be involved in anti-viral response, namely NK and T cells. Indeed, interferon (IFN) response and the activation of NK cells are the first-line defense against viruses, and the failure of these mechanisms activates adaptive cellular response [4,15]. The latter is mostly based on Tc cells that recognize viral antigens presented by human leukocyte antigen (HLA) class I molecules on the surface of infected cells. In order to better understand immunological status of a B-cell deficient patient and decipher which of the immune system features could be vital for relatively mild course of COVID-19 and complete remission, we performed a unique flow cytometry analysis of T and NK cells based on ten 16-antibody panels.

We are aware of only one report describing T cell-mediated responses in an immuno-compromised patient managed with obinutuzumab, a case with prolonged SARS-CoV-2 viremia following chemotherapy and obinutuzumab for follicular lymphoma [12]. As was the case in our case study, the patient did not produce anti-SARS-CoV-2 antibodies. However, two courses of remdesivir and subsequent CP induced sustained virological control of COVID-19. There was also a mild response of CD4+ and CD8+ T cells against single peptides of the envelope, membrane, and nucleocapsid proteins of SARS-CoV-2. The responses were measured as the percentage of cells producing IFN-γ after stimulation with the viral peptides. Interestingly, clinical improvement was associated with an increase, and clinical recurrence with a decrease of Tc (CD8+) and NK cell counts. The authors suggested that a decrease in the CD4+ to CD8+ T cell ratio during COVID-19 can be predictive for clinical improvement [12]. However, these alterations seem to result from a significant increase in Tc (CD8+) cell counts, but not a decrease in Th (CD4+) cell numbers, which were also increased during clinical improvement. 

These results correspond with our data. Our patient had normal counts of Tc and NK cells before and after the infection, and Tc cells in these periods accounted for 67.0% and 62.5% of peripheral blood lymphocytes, respectively. Interestingly, the patient also showed antigen-specific T cell activation and proliferation. To the best of our knowledge, this is the first study demonstrating the TCR-specific oligoclonal response of T cells in COVID-19. Among the 24 studied TCR Vβ families of Tc cells, the most striking difference between the patient and the HD was found for the frequency of Tc clones expressing TCR Vβ2 (16.7% vs. 3.7%, respectively). Additionally, these clones showed markers of activation and prolonged TCR stimulation. There was a remarkable difference in the proportion of TEMRA cells among TCR Vβ2 Tc cells (87.3% in the patient vs. 16.6% in the HD). TEMRA cells are considered a subset of TEM cells that re-express CD45RA after antigen stimulation and may be implicated in protection from infections with RNA viruses (e.g., dengue virus) [16]. TEMRA cells were shown to have higher levels of clonal expansion and consist of virus-specific cells [16]. High levels of TEM and TEMRA cells within the Tc cell population were also shown to be associated with response to antiretroviral therapy in HIV-1-infected patients [17]. In addition, Northfield et al. demonstrated that CD8+ TEMRA cells have superior antiviral activity and ability for long-term control of HIV-1 infection [18]. Until now, there has been only one report on a single B-cell depleted lymphoma patient who experienced uncomplicated COVID-19 and showed increased proportions of CD8+ TEMRA cells. In the same study healthy age- and gender-matched individuals who recovered from COVID-19 were also characterized by increased numbers of CD8+ TEMRA cells. Interestingly, severe course of SARS-CoV-2 infection in B-cell deficient patients was associated with significantly lower proportions of TEMRA cells within T CD8+ cell subset [19]. These results suggest that CD8+ TEMRA cells may be responsible for the control of SARS-CoV-2 and simultaneously can serve as a marker of positive clinical outcome. Our data support this hypothesis and add novel information required for understanding of anti-SARS-CoV-2 T cell response. In the current study, we identified a clonal expansion of TCR Vβ2 CD8+ T cells highly enriched (83.7%) in TEMRA cells in a B-cell depleted patient with a mild course of COVID-19. Notably, TCR Vβ2 Tc cells from our patient expressed markers of strong activation, as compared with the same cell popu-lation derived from HD who had no history of SARS-CoV-2 infection. Nearly all (82.1%) of TCR Vβ2 Tc cells from the patient produced perforin and expressed 4-1BB, a marker of TCR-mediated activation, whereas in the HD these cells accounted for only 20.7%. Moreover, 40.8% of TCR Vβ2 cells in the patient simultaneously expressed IFN-γ and IL-10, postulated as a mechanism preventing autoimmunity and immunopathology during chronic infections [20]. Altogether, these data suggest that proliferation and activation of TCR Vβ2 Tc cells in our patient were manifestations of the efficient antigen-specific antiviral response. It is clear that TCR Vβ2 Tc cells comprise of various TCR clones which share TCR Vβ2 chain, thus further studies are required to fully determine SARS-CoV-2 specific T cell clones.

In the current study we also screened our patient for distribution and expression of cytotoxic effector molecules in five different NK cell subsets. The patient showed normal total NK cell counts. However, detailed analysis revealed an increased frequency of the CD56-CD16+ NK cell subset. NK cells with this phenotype have been previously linked to chronic viral infections [21]. Thus, increased numbers of these cells may also accompany persistent but undetectable viral replication in SARS-CoV-2-infected individuals, which is in accordance with the clinical status of our patient. Previous studies on other RNA viruses, namely HIV-1, also showed that chronic infections with high viral loads are associated with decreased frequencies of CD56dimCD16+ NK cells and simultaneous increase in CD56-CD16+ cell counts [21]. We observed a similar response in our patient. However, there is a lack of information on frequency and characteristics of CD56-CD16+ NK cells in SARS-CoV-2-infected individuals, and our study is the first that focused on this phenomenon in COVID-19. Interestingly, 99% of CD56-CD16+ NK cells in our patient were positive for granzyme A, while in HD, cells expressing this molecule were significantly less numerous (mean 85%). Noteworthy is that granzyme A was shown to have multiple noncytotoxic functions that inhibit viral replication and spread. Granzyme A was also suggested to be the bridge between the innate and adaptive immune system by enhancing release of type I IFNs and maturation of plasmacytoid, as well as conventional dendritic cells via the TLR9-MyD88 pathway. Thus, granzyme A was shown to induce large T cell responses after antigen exposure [22,23]. These data suggest that granzyme A positive CD56-CD16+ NK cells may contribute to a mild course of SARS-CoV-2 infection. However, further studies on this topic are needed. 

The prolonged COVID-19 and virus shedding in our patient presumably resulted from the lack of B cells and low counts of Th cells. However, this deficiency was compensated to some extent by robust activation of Tc and NK cells. Thus, a relatively mild course of infection might have resulted from adequate cellular native (NK) and adaptive (Tc) immune response. In combination with remdesivir and CP, these features led to complete clinical resolution, despite a lack of seroconversion. Interestingly, the patient was also negative for anti-SARS-CoV-2 antibodies after CP administration. This could at least partially result from the immediate binding of these antibodies to viral antigens. Another more likely explanation is an unspecific binding of antibodies with Fc fragment receptors present on the surface of innate immune cells, primarily neutrophils (the most abundant cells in peripheral blood) and NK cells. Subsequently, these immune cells coated with anti-SARS-CoV-2 antibodies could specifically bind via free Fab fragments of the surface-bound antibodies with the virus-infected cells, leading to complete clinical resolution and a negative qRT-PCR result.

Co-administration of remdesivir and CP in our patient makes it difficult to assess the contribution of either method to the viral clearance. However, recently published data showed that in three out of four immunocompromised patients treated with consecutive administration of remdesivir and CP, the viral eradication was only achieved with the latter. The efficacy of CP was confirmed recently in a series of 17 patients with profound B-cell lymphopenia and protracted COVID-19 disease [24]. Hence, the CP appears to be an effective and safe approach in such patients and may be crucial for viral clearance, even after the first 10 days.

The main limitation of our study is that it describes a single individual whose immune response may not be broadly generalizable to other B-cell depleted patients. However, as compared with age- and gender-matched HD studied simultaneously, our study clearly demonstrates the uniqueness of the immune status of the patient and helps to understand which elements of his immune system helped him to survive this life-threate-ning infection.

It is in this way that our study sheds new light on the role of cellular immune responses in the clinical outcome of COVID-19. The case of our patient indicates that the simultaneous use of remdesivir and CP for protracted COVID-19 in B-cell deficient patients may lead to viral clearance when activation of NK cells and antigen-specific Tc cells is induced. These results also contradict the general conviction that the beneficial effect of remdesivir and CP can only be achieved at the beginning of the infection. We believe that this case report will facilitate further research on antigen-specific response to the virus and may contribute to new strategies for monitoring and clinical decision-making in high-risk patients with COVID-19.

## 4. Materials and Methods

Three weeks after the negative qRT-PCR test, the patient underwent profound immune profiling, focusing on the clonal expansion of Th, Tc, and regulatory T cells (Tregs), and the activation status of T, NK, and B cells. Simultaneously, for comparison, peripheral blood mononuclear cells from three male healthy HD without a history of COVID-19 and matched for age were collected and analyzed using flow cytometry. The patient and HD provided written consent, and the study was approved by the Independent Bioethics Committee for Scientific Research at Medical University of Gdańsk. 

SARS-CoV-2 RNA was quantified with qRT-PCR (VIASURE Real-Time PCR; Certest Biotec S.L., Zaragoza, Spain). The test is designed for the amplification of a conserved region of ORF1ab and N genes for SARS-CoV-2, and its detection limit is ≥10 RNA copies per reaction for ORF1ab and N genes. The antigen tests were performed using the SARS-CoV-2 Rapid Antigen Test (Roche Diagnostics, Basel, Switzerland). Roche Elecsys anti-SARS-CoV-2 tests were used for the qualitative detection of anti-SARS-CoV-2 antibodies. The assay uses a recombinant protein representing the nucleocapsid (N) antigen in a double-antigen sandwich assay format, which favors detection of high-affinity antibodies (IgM and IgG) against SARS-CoV-2. Analysis of immune cells and detailed characterization of their phenotype were performed with the following 16-antibody panels for flow cytometry purchased from BD Biosciences (San Jose, CA, USA), Thermo Fisher Scientific (Waltham, MA, USA), BioLegend (San Diego, CA, USA), and Beckman Coulter (Brea, CA, USA): (1): CD4, CD8, CD25, CD45RA, CD62L, 4-1BB, CD40L, KIR3DL1, PD-1, IL-10, Ki67, IFN-g, FoxP3, perforin, and TCR-detection kit (based on two fluorophores); (2): CD3, CD16, CD56, perforin, granzyme A, IFN-γ, KIR3DL1, TLR-2, TLR-4, TLR-9, CD94, CD314, PD-1, PD-L1, KIR2DL4, and CD69; and (3): CD19, CD27, CD38, CD24, CD5, HLA-DR, CD25, CD138, CD20, IgM, IgD, CD1d, PD-1, CD21, CD10, and CD40.

The first panel was repeated eight times, as it was combined with staining with Beta Mark TCR Vbeta Repertoire Kit (Beckman Coulter, Brea, CA, USA) designed for the quantitative determination of 24 families of the TCR Vβ repertoire of human T cells [25]. The kit is composed of eight sets of mixtures of fluorochrome-conjugated antibodies against TCR Vβ corresponding to 24 different specificities (24 TCR Vβchain families; about 70% coverage of normal human TCR Vβ repertoire). The kit enables the detection of three different Vβ chains in the same tube using three monoclonal antibodies (mAb) with only two fluorophores. The first mAb is FITC-conjugated, the second is PE-conjugated, and the third is conjugated with both PE and FITC. Thus, this approach reduces the number of samples for analysis from 24 to 8. Simultaneously, naive (TN), central memory (TCM), effector memory (TEM), and effector memory T cells that re-expressed CD45RA (TEMRA) were characterized by the following phenotypes: CD45RA + CD62L+, CD45RA-CD62L+, CD45RA-CD62L-, and CD45RA + CD62L-, respectively [26]. 

Analysis of these T cell subsets is crucial for identification of antigen-specific immune response, notably in the context of the persistent SARS-CoV-2 infection. Prolonged contact with the virus results in priming of pathogen-specific TN cells leading to their differentiation into memory subsets [27]. Thus, alterations in proportions of TN, TCM, TEM and TEMRA cell subsets bearing the particular TCR Vβ reflect the antigen-specific response. Based on the relative expression of the surface markers CD16 and CD56, human NK cells can be divided into five subsets. Thus, in the current study we have identified and the following NK cell populations: CD3-CD56dimCD16+, CD3-CD56brightCD16-, CD3-CD56dim CD16-, CD3-CD56brightCD16dim, and CD3-CD56-CD16+ (gating strategy in Appendix A), as described previously [4], among which CD3-CD56rightCD16- and CD3-CD56dimCD16+ are dominant in healthy individuals [6]. The blood samples from HD were analyzed using the same flow cytometry method and were used for comparison as a reference. All values in this group are presented as means. The data were analyzed with a BDFACSAria Fusion 5 laser flow cytometer and sorter (BD Biosciences, San Jose, CA, USA). Statistical plots were performed with STATISTICA 13.3 (StatSoft, Cracow, Poland) software, and Figure 1 was performed with BioRender (Toronto, ON, Canada) graphical software. 

## 5. Conclusions

Simultaneous use of remdesivir and CP for protracted COVID-19 in B-cell depleted patients may lead to viral clearance when activation of NK cells and antigen-specific Tc cells is induced. 

## Figures and Tables

**Figure 1 ijms-22-10934-f001:**
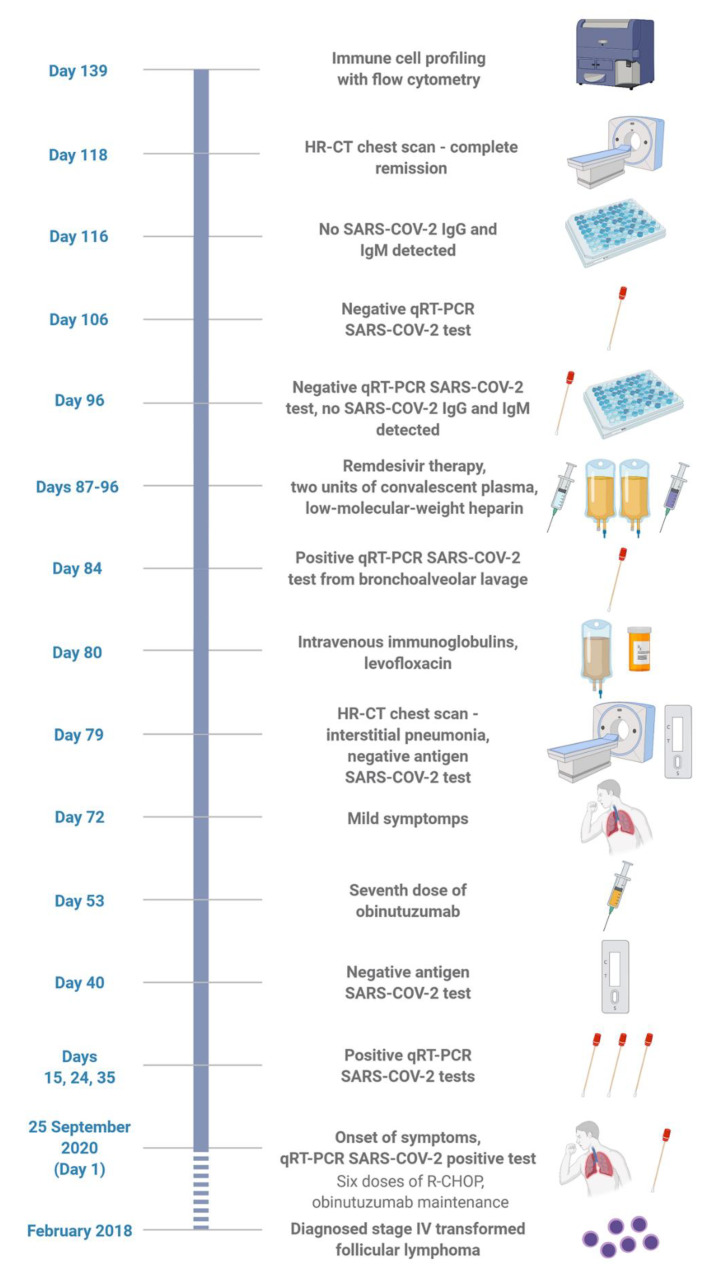
Clinical timeline (until day 139 after diagnosis of COVID-19). The figure was performed with BioRender (Toronto, ON, Canada) graphical software.

**Figure 2 ijms-22-10934-f002:**
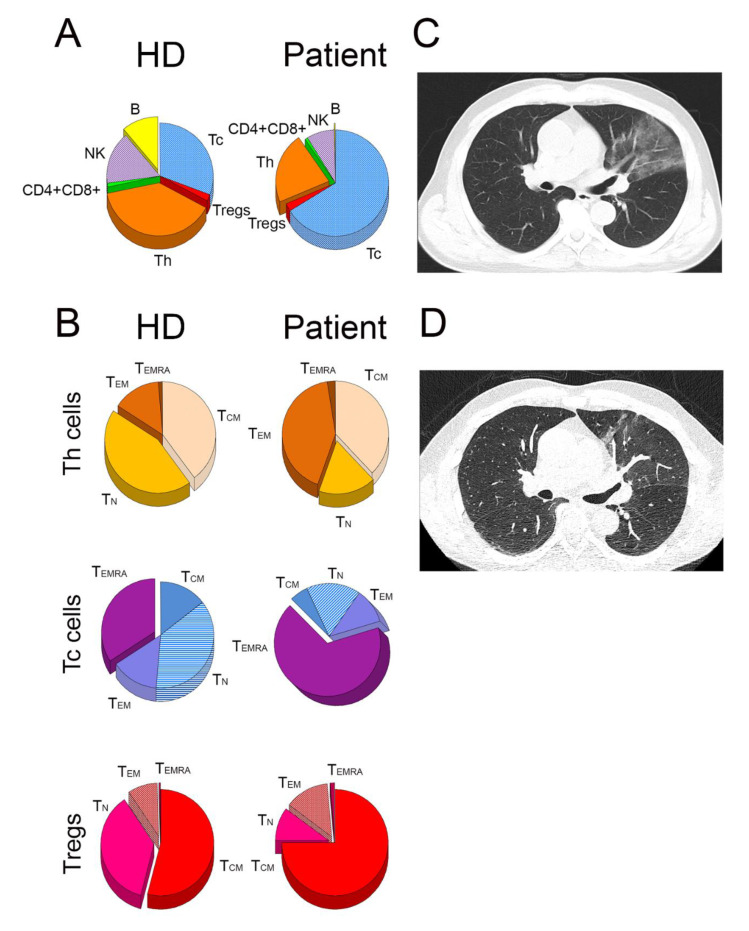
(**A**) Depicts the distribution of the following lymphocyte subsets: Tc, Th, Tregs, CD4 + CD8+ (double positive) T cells, B and natural killer cells in the healthy donors (HD) and in the patient. (**B**) Depicts the proportions of naive (TN), central memory (TCM), effector memory (TEM), and effector memory T cells that re-expressed CD45RA (TTEMRA) within Th, Tc, and Treg cell populations. The right panel depicts the high-resolution computed tomography chest scans at day 79 (**C**; interstitial pneumonia) and at day 118 (**D**; complete remission).

**Figure 3 ijms-22-10934-f003:**
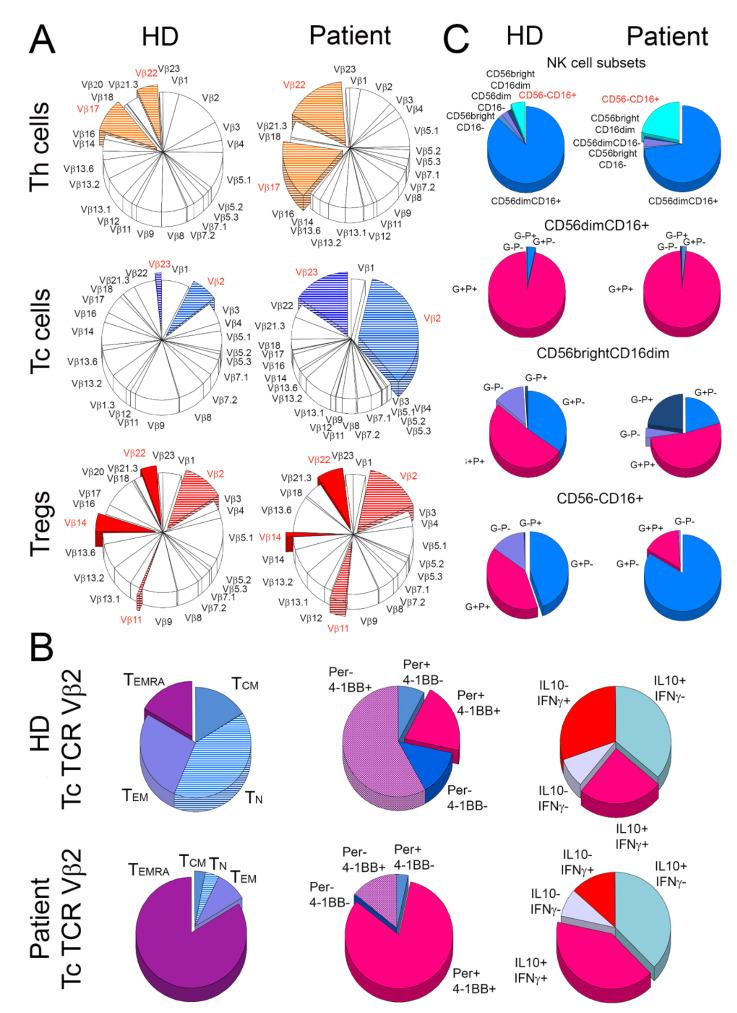
Oligoclonal expansion and activation of T cells in the patient. The figure depicts (**A**) mean frequency distribution of 24 T cell receptor (TCR) Vβ families of T cells in three healthy donors (HD) and the patient (Patient). The most striking percentage differences between the patient and the control donors are marked with colors. (**B**) Shows characteristics of cytotoxic T (Tc) cells from TCR Vβ2 family in healthy donors (HD) and the patient (Patient). For both the patient and HD, the frequency of naive (TN), central memory (TCM), effector memory (TEM), and effector memory Tc cells that re-expressed CD45RA (TTEMRA) were compared (1st column). In addition, TCR Vβ2 Tc cells were compared for the proportions of perforin (Per) and 4-1BB (activation marker) positive (+) and negative (−) cells (2nd column) and for the distribution of cells positive (+) and negative (−) for IL-10 and IFN-g (3rd column). The right panel (**C**) shows the distribution of five subsets of natural killer (NK) cells (1st row) and then cells positive (+) and negative (−) for perforin (P) and granzyme A (G) within CD56dimCD16+ (2nd row), CD56brightCD16dim (3rd row), and CD56-CD16+ (4th row) NK cell subsets.

**Table 1 ijms-22-10934-t001:** Case reports of COVID-19 infection course in immunocompromised patients.

Author	Age; Sex	Diagnosis	Reason for B-Cell Immune Deficit	Infection Duration (Days)	ARDS/Multiorgan Involvement	Efficacy of Remdesivir (Treatment Duration)	Effect of Convalescent Plasma	Seroconversion
Helleberg et al. [9]	50+; M	CLL	6 cycles of CFR	60	no	Reduced viral load after 2nd attempt (10 days each)	Viral clearance (2 infusions)	no
Avanzato et al. [2]	71; F	CLL	Long-term CLL therapy; hypogammaglobulinemia	105	no	NA	Viral clearance (2 infusions)	no
Martinot et al. [10]	76; F	CLL	4 cycles of RB	41	no	Short-term clinical improvement	Viral clearance (4 infusions)	no
Camprubí-Ferrer et al. [11]	37; F	Relapsed FL	3 cycles of R-ESHAP	63	no	Viral clearance after 2nd attempt (10 days each)	NE ^1^	no
Malsy et al. [12]	53; F	FP	O-CHOP; obinutuzumab maintenance	94	no	Clinical improvement (10 days, 5 days)	Viral clearance ^2^ (2 courses of 6 units)	no
Buckland et al. [13]	31; M	XLA	Primary immunodeficiency	64	no	Short-term clinical improvement after 1st course, viral clearance after 2nd attempt (10 days each)	NE ^1^	no
Baang et al. [14]	60; M	Refractory MCL	COP + 2 consecutive B-cell directed antibodies	156 ^3^	no	Short-term clinical improvement after each attempt (10 days each) ^3^	Short-term clinical improvement ^3^ (2 infusions)	yes
Current case	69; M	DLBCL	6 cycles of R-CHOP; obinutuzumab maintenance	96	no	Clinical improvement, viral clearance (10 days)	Clinical improvement, viral clearance (2 infusions)	no

CLL, chronic lymphocytic leukemia; CFR, cyclophosphamide, fludarabine, rituximab; ARDS, acute respiratory distress syndrome; FL, follicular lymphoma; R-ESHAP, rituximab, etoposide, cisplatin, cytarabine, methyl-prednisolone; NA, not applicable; O-CHOP, obinutuzumab, cyclophosphamide, doxorubicin, vincristine, prednisone; MCL, mantel cell lymphoma; XLA, X-linked agammaglobulinemia; RB, rituximab, bendamustine; DLBCL, diffuse large B-cell lymphoma; R-CHOP, rituximab, cyclophosphamide, doxorubicin, vincristine, prednisone; ^1^ NE, not evaluable; convalescent plasma administered after viral clearance; ^2^ convalescent plasma combined with the second course of remdesivir; ^3^ remdesivir and convalescent plasma were administered concurrently.

## Data Availability

All data needed to evaluate the conclusions in the paper are present in the main text or the Appendix A. The raw data are available upon request from N.M.T. after signing a confidentiality agreement.

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
