# Peer review of "Successful Treatment of Persistent SARS-CoV-2 Infection in a B-Cell Depleted Patient with Activated Cytotoxic T and NK Cells: A Case Report"

_ijms, 2021, doi:10.3390/ijms222010934_

Round 1

Reviewer 1 Report

The case study report titled "Successful treatment of persistent SARS-CoV-2 in a B-cell depleted patient with activated cytotoxic T and NK cells: a case report" by Jassem J et. al. is discussing the immune response of a single B cell depleted lymphoma patient to a mild to moderate SARS-COV-2 infection.  The authors had studied NK and TCR-specific T cell oligoclonal response in the patient after clearing the virus. They concluded that NK and antigen specific Tc cell response are sufficient to clear the virus. This conclusion doesn't add to the current knowledge of the immune response against SARS-COV-2. In addition, the authors hypothesise that the simultaneous use of remdesivir and CP at protracted COVID-19 in B cell-depleted patients may lead to viral clearance. Although they don't show any data, comparisons or even analysis supporting this hypothesised conclusion, it is not adding to our current knowledge of the infection.

The authors found that viral clearance was likely attributed to the robust expansion and activation of TCR Vβ2 CD8+ cytotoxic T cells and CD16+CD56- NK cells. The analysis was totally based on phenotypes without any functional studies on a single patient which doesn't render any strength to the conclusion. The discussion is very weakly written not well justifying the results. The comparison was mainly based on healthy donors with normal cellular repertoire, this is an unfair comparison especially when based only on phenotypic analysis as B cells will definitely consume a percentage that is absent in the presented patient. Instead it would have been more valid to compare the patient with other immunocompromised patients who experienced other viral infections than SARS-COV-2 demonstrating the absence of TCR Vβ2 CD8+ cytotoxic T cell expansion, if true.

Analysed NK cell population is not well discussed regarding their different role and function in viral infection. Double error in the results and methods indicating a non existing population "CD3-CD56rightCD56dim" line 121 and line 402. NK cell function is not analysed to support the hypothesised conclusion. NK-T cells are not analysed and completely disregarded

No statistical analysis found which is conceivable as it is only one patient yet the authors state that the Statistical analysis was performed with STATISTICA 13.3 (StatSoft, Cracow, 408 Poland) software!

Doubtful results: Immune profiling started on day 139 which is quiet a long period during which we don't know if the patient carried SARS-COV-2 all through or was re-infected as the clearance was demonstrated by a single nasopharyngeal antigen assay on day 40 which is not enough.

Author Response

1. The case study report titled "Successful treatment of persistent SARS-CoV-2 in a B-cell depleted patient with activated cytotoxic T and NK cells: a case report" by Jassem J et. al. is discussing the immune response of a single B cell depleted lymphoma patient to a mild to moderate SARS-COV-2 infection.  The authors had studied NK and TCR-specific T cell oligoclonal response in the patient after clearing the virus. They concluded that NK and antigen specific Tc cell response are sufficient to clear the virus. This conclusion doesn't add to the current knowledge of the immune response against SARS-COV-2.

We would like to thank the Reviewer for the comments. We understand the Reviewer point of view. However, we do not claim that NK and antigen specific Tc cell response are sufficient to clear SARS-CoV2. We just stated that the activation of these cells might be an important factor required for survival of SARS-CoV2 infection. We would like also to kindly point out that this is the first study where Th, Tc and Treg TCR clones were compared for proportions and activation status in immunocompromised patient affected by SARS-CoV2, as well as in healthy individuals. This is also the first study in B cell deficient individual where NK cells and their activation was studied in such details (we studied 5 NK cell subsets and checked their activation status).

2. In addition, the authors hypothesise that the simultaneous use of remdesivir and CP at protracted COVID-19 in B cell-depleted patients may lead to viral clearance. Although they don't show any data, comparisons or even analysis supporting this hypothesised conclusion, it is not adding to our current knowledge of the infection.

We would like to notice that our patient had positive qRT-PCR test at day 84 of the infection (since the day 1).  Then, remdesivir and CP were administered on days 86-97 to our patient. Immediately after this treatment the patient was SARS-CoV2 negative in qRT-PCR test (day 96). Negative result of the diagnostic test and symptom resolution are considered to be the prove of viral clearance. The novelty of this study is not the therapeutic regimen used, but the time if its implementation (day 86 of the infection) and first of all- the detailed  immunological screen of the patient immune response with TCR clone analysis for basic T cell populations.

3. The authors found that viral clearance was likely attributed to the robust expansion and activation of TCR Vβ2 CD8+ cytotoxic T cells and CD16+CD56- NK cells. The analysis was totally based on phenotypes without any functional studies on a single patient which doesn't render any strength to the conclusion. The discussion is very weakly written not well justifying the results.

We would like to notice that we not only show the cell phenotype but also tested intracellular cytokine and functional molecule synthesis. Please see Figure  3B where perforin, IL-10, IFN-gamma, as well as activation marker 4-1BB were analysed or Figure 3C, where perforin and granzyme expression was tested. Measurement of cytokine production and expression of activation markers is in fact testing of cell function. Please notice that ELISA or ELISPOT assays used for assessment of cell function also measure cytokine levels. The only difference between these tests and our approach is that we measured cytokine expression inside the cells, while ELISA or ELISPOT measure the cytokine levels after cytokine secretion. 

4. The comparison was mainly based on healthy donors with normal cellular repertoire, this is an unfair comparison especially when based only on phenotypic analysis as B cells will definitely consume a percentage that is absent in the presented patient. Instead it would have been more valid to compare the patient with other immunocompromised patients who experienced other viral infections than SARS-COV-2 demonstrating the absence of TCR Vβ2 CD8+ cytotoxic T cell expansion, if true.

We understand the Reviewer point of view. At the same time we would like also to notice that our paper is the case study. By definition the case study is focused on 1 patient and discusses the single case for the purpose of understanding of a larger population or -as it is in case of our case study- understanding of a certain mechanism/phenomenon. In the paper we decided to present immune status of the healthy age and gender matched individuals to demonstrate the deviation between the immune status of our patient and the immune status of healthy individuals. Our intension of such data presentation was to underline the uniqueness of the immune status of the patient and help to understand which elements of his immune system could help him to survive this life threatening disease.

The aim of the current study was to present a detailed immune characteristics of the B cell deficient patient who suffered from persistent but not severe SARS-CoV2 infection and overall recovered without further complications. With this case study we aimed to put attention of the researchers to the immune elements which can be crucial for SARS-CoV2 survival.

We also understand the Reviewer request for analysis of blood samples of other immunocompromised patients who experienced other viral infections than SARS-COV-2 to demonstrate the absence of TCR Vβ2 CD8+ cytotoxic T cell expansion. However, please pay attention that TCR Vβ2 CD8+ cytotoxic T cells are not a single clone but in fact a family of TCR clones, as TCRs are composed not only of beta but also alpha chains. Within TCR Vβ2 family multiple various T cell clones exist. Thus, this family may contain T cells against SARS-CoV2 and also other viruses and infectious agents. Thus, the exact SARS-CoV2 specific clones need to be determine in the future studies and it was not the aim of the current study. With this study we just shed light on a TCR family that might be important for SARS-CoV2 eradication. It is also possible that some cross reactivity occurs between SARS-CoV2 specific T cells and T cells specific to other viruses. Therefore, recruitment of a group of immunocompromised patients infected with other viruses could not give  the response to the Reviewer question.

We would like also to bring to the Reviewer attention that comparing immune responses of our patient with other patients with the same type of lymphoma but infected with other viruses would be very challenging in terms of data interpretation. For this kind of study we would need perfectly matched group for age, gender, treatment regimen and infected with a virus at the same time point during the therapy as our patient. All these factors affect immune response. Unfortunately recruitment of such group is not possible for a single medical centre and would be even a big challenge for a multicentre study.

Taking all these information together, we want to underline that the aim of this case study was characterize deeply the immune status of the B cell deficient patient to show its uniqueness and shed light on the mechanisms involved in COVID19 survival in B cell deficient patients.

5. Analysed NK cell population is not well discussed regarding their different role and function in viral infection. Double error in the results and methods indicating a non existing population "CD3-CD56rightCD56dim" line 121 and line 402. NK cell function is not a0nalysed to support the hypothesised conclusion. NK-T cells are not analysed and completely disregarded

We would like to thank the Reviewer for pointing out these typos. We have already corrected them in the text (please see line 124 and 404 in the corrected version of the Manuscript with change highlighted). The analysis of NK cells is performed at the Figure 3C. Please notice that the upper panel shows various distribution of 5 NK cell subsets, while the lower panels depict production of perforin and granzyme in CD56dimCD16+, CD56brightCD16dim and CD56-CD16+ NK cell subsets. Please also see the paragraph: 2.3 Alterations in the distribution of NK cells and their activation status in the Results section (lines 121-130 in the corrected version of the Manuscript with changes highlighted). We also discuss the significance of these results in the Discussion section (please see lines 319-327 130 in the corrected version of the Manuscript with changes highlighted). Our data show long term and robust activation of NK cells in the patient reflected by increased numbers of CD3-CD16+CD56- NK cells (signature of chronic viral infection; please see the paper PMID: 25177322) positive for granzyme but negative for perforin. Please notice that this is a case study and relatively brief report. Thus, each section is concise according to the Journal policy. We admit that we did not analyse NK-T cells. However, this was not the aim of the current study. We also did not analyse γδT cell, DCs and Monocytes. However, this is usually not possible to focus on all immune mechanisms in a single study. We are convinced that the Reviewer understands this issue.

6. No statistical analysis found which is conceivable as it is only one patient yet the authors state that the Statistical analysis was performed with STATISTICA 13.3 (StatSoft, Cracow, 408 Poland) software!

We would like to apologize for this imprecision. We have used the STATISTICA 13.3 software to perform statistical charts. We made this clear in the current version of the paper. Please see the line 410 in the corrected version of the Manuscript with changes highlighted.

7. Doubtful results: Immune profiling started on day 139 which is quiet a long period during which we don't know if the patient carried SARS-COV-2 all through or was re-infected as the clearance was demonstrated by a single nasopharyngeal antigen assay on day 40 which is not enough.

The flow cytometric analysis was performed on day 139, that was 43 days after the negative qRT-PCR test. However, please bear in mind that T cells are long-living cells and any immune challenge leaves its immune footprint in T cell subset. The contact with the pathogen or vaccine generates the immune memory. Prolonged stimulation with the antigen is also responsible for generation of TEMRA cells that was observed in our patient.

NK cells were previously thought to have a relatively short lifespan. However, currently is known that after viral infection, long-lived NK cells are generated and reside for months in lymphoid tissues, as well as nonlymphoid sites, and can rapidly mount protective secondary responses when virus is reencountered (please see the paper published in Nature in 2009 PMID: 19136945).

We can not agree that the patient was reinfected. Please see in the text (lines 70-78 in the revised version of the Manuscript with changes highlighted) on day 79, a high-resolution computed tomography (HRCT) of the chest showed left-side lobular pneumonia with ground-glass opacities and slight pleural effusion (CORADS 3; Figure 2C), but at the same time the Roche SARS-CoV-2 antigen test was negative, while qRT-PCR from bronchoalveolar lavage tested positive for SARS-CoV-2. It has been reported before that qRT-PCR is more accurate method for evaluation of SARS-CoV2 infection than antigen tests. Thus, qRT-PCR is considered a gold standard in terms of SARS-CoV2 diagnostic.

We hope that with these responses we were able to dissipate all the doubts of the Reviewer and that the paper can be accepted for the publication.

Reviewer 2 Report

The paper titled “Successful treatment of persistent SARS-CoV-2 in a B-cell 2 depleted patient with activated cytotoxic T and NK cells: a case 3 report” is a very interesting work take into account the pandemic situation in the world. Moreover, it interest is due too by their results because in this paper authorssuggests that B cell- depleted patients may effectively respond to anti-SARS-CoV-2 treatment when NK and antigen- specific Tc cell response is induced.

There are only one minor points.

  • Author give a compressive analysis of lymphocyte subsets in figure 1 and T cells in figure 2. However, this analysis is performed three weeks after negative qRT-PCR test, and I think that it will be very interesting to can perform this analysis after the firs positive SAR-COV2 test, in day 40 and after the seventh dose of obinutuzumad in order to see if the lymphocyte and T cellssubsets were altered by this conditions and how is this variation.

Author Response

We would like to thank the Reviewer for the comments and time spend on evaluation of our paper. We have done our best to respond to all the questions and issues raised.

The paper titled “Successful treatment of persistent SARS-CoV-2 in a B-cell 2 depleted patient with activated cytotoxic T and NK cells: a case 3 report” is a very interesting work take into account the pandemic situation in the world. Moreover, it interest is due too by their results because in this paper authors suggests that B cell- depleted patients may effectively respond to anti-SARS-CoV-2 treatment when NK and antigen- specific Tc cell response is induced.

There are only one minor points.

Author give a compressive analysis of lymphocyte subsets in figure 1 and T cells in figure 2. However, this analysis is performed three weeks after negative qRT-PCR test, and I think that it will be very interesting to can perform this analysis after the first positive SAR-COV2 test, in day 40 and after the seventh dose of obinutuzumab in order to see if the lymphocyte and T cell subsets were altered by this conditions and how is this variation.

We thank the reviewer for the overall positive appraisal of our work and raising this point, which is worthy of clarification. Indeed, the analysis of lymphocyte subsets performed at these time points would provide additional information. Unfortunately, immunological assessment of this case was not planned a priori, and profound immune profiling was performed only after the eradication of SARS-Cov-2 infection had been achieved.

Reviewer 3 Report

The case report by Jassem et al. describes a patient with COVID-19 and concurrent lymphoma on B-cell repletion therapy. This is an interesting case report with the extensive investigation in lymphocyte subsets. We would want to see some clarification before publication.

  1. (Case – Introduction) Why did you empirically treat a patient with IVIG? It is sometimes used for patients with CVID but not for hematologic patients; the authors would want to mention supporting evidence on it in discussion at least.

  1. (Case presentation) The positive qRT-PCR finding on bronch lavage, would it be just residual of the recent infection, not viable and clinically meaning full viral load? The authors noted levoflo and IVIG were ineffective, but could it also be just that levofloxacin did not cover causative bacterial pathogens?

Author Response

We would like to thank the Reviewer for the comments and time spent on evaluation of our paper. We have done our best to respond to all the questions and issues raised. We have also made required clarifications in the text.

The case report by Jassem et al. describes a patient with COVID-19 and concurrent lymphoma on B-cell repletion therapy. This is an interesting case report with the extensive investigation in lymphocyte subsets. We would want to see some clarification before publication.

  1. (Case – Introduction) Why did you empirically treat a patient with IVIG? It is sometimes used for patients with CVID but not for hematologic patients; the authors would want to mention supporting evidence on it in discussion at least.

We thank the Reviewer for the overall positive appraisal of our work and raising this issue. The decision on empirical treatment with IVIG was based on local standard guidelines to administer iv IgG in all patients with severe recurrent respiratory infections secondary to hypogammaglobulinemia.

  1. (Case presentation) The positive qRT-PCR finding on bronch lavage, would it be just residual of the recent infection, not viable and clinically meaning full viral load? The authors noted levoflo and IVIG were ineffective, but could it also be just that levofloxacin did not cover causative bacterial pathogens?

We thank the reviewer for raising this point. The diagnosis of COVID-19 reinfection was taken by the multidisciplinary team, considering the clinical picture, the positive result of PCR assay in BAL, and the negative results of comprehensive microbiological tests (cultures for aerobic, anaerobic bacteria, fungi and TB, and PCR multiplex for 22 viral and 33 bacterial pathogens). Hence, there was no indication for second-line antibiotic therapy. Dramatic clinical improvement after the introduction of remdesivir and convalescent plasma confirmed ex juvantibus the diagnosis of COVID-19. Thus, altogether, we had strong arguments that SARS particles detected by PCR in BAL were not residual but causative and that the probability that undetected other bacterial pathogen caused pneumonia was very unlikely.

In the current version of the manuscript we have added more details on microbiological testing. Please see page 3 lines 73-78 in the corrected version of the Manuscript with changes highlighted.

Round 2

Reviewer 1 Report

The authors had provided detailed and profound response to the reviewers comments. However the text had not been sufficiently modified. It is conceivable that it is a brief case study that cannot address all aspects yet a case study either presents a rare case to consider or novel information. The novelty in the current text presentation is not well emphasised. Comparing cellular percentages with healthy donors in a B cell deficient case weakens the study. The authors should at least compare with the literature of immune compromised patients' response to viral infection and demonstrate what is unique about the immune response of their patient. The current form of the case presentation demonstrates a natural immune response to viral infection in a B cell deficient patient. Yet the authors insist that it is a unique response while this is not shown by comparing cellular percentages to healthy donors. The authors had elegantly demonstrated case reports of COVID-19 infection course in immunocompromised patients in Table1, a similar table would demonstrate their unique result when well discussed.

Author Response

  1. The authors had provided detailed and profound response to the reviewers comments. However the text had not been sufficiently modified. It is conceivable that it is a brief case study that cannot address all aspects yet a case study either presents a rare case to consider or novel information. The novelty in the current text presentation is not well emphasised.

We would like to thank the Reviewer for the time spent on our paper evaluation and all advices given. We have done our best to modify the manuscript accordingly. Please see the current version of the Manuscript with changes highlighted. We have emphasized the novelty of the study (lines 287-289; 293-297), notably T cell analysis (323-362), as well as NK cell screen (363-387) in the Discussion section. We hope the modifications are satisfactory.

  1. Comparing cellular percentages with healthy donors in a B cell deficient case weakens the study. The authors should at least compare with the literature of immune compromised patients' response to viral infection and demonstrate what is unique about the immune response of their patient. The current form of the case presentation demonstrates a natural immune response to viral infection in a B cell deficient patient.

We would like to thank the Reviewer for this advice. We have screened again the available literature on the topic. We did not find a study that recruited B cell depleted patient/patients and would present similar deep flow cytometry analysis of immune response as we showed in the current paper. However, we have done our best to search again through the literature and find at least some elements that could be compared. Please see the Discussion section in the revised version of the Manuscript  (lines 330-345). At this point we would like also to emphasize that the main aim of our study was not to illustrate the uniqueness of immune response in SARS-CoV2 infected patient, but to show which immune element might be important for COVID19 survival of B- cell deficient patient. We believe that our study will be the starting point for future research on new strategies for monitoring and decision making during treatment of high risk patients infected with SARS-CoV2. Infections with different viruses might induce similar immune responses. However, in the current study the most important was to indicate if the particular response has clinical meaning in monitoring of patient outcome during SARS-CoV2 infection. We believe that with revised Manuscript we were able to indicate and emphasize the significance and possible impact of CD8+ TEMRA cells and granzyme A positive CD56-CD16+ NK cells on the clinical outcome of COVID19. 

  1. Yet the authors insist that it is a unique response while this is not shown by comparing cellular percentages to healthy donors. The authors had elegantly demonstrated case reports of COVID-19 infection course in immunocompromised patients in Table1, a similar table would demonstrate their unique result when well discussed.

We would like to thank the Reviewer for this suggestion. We have added two supplementary tables that clearly present ccharacteristics of TCR Vb2+CD8+ T cells in the patient and healthy donors (please see Table S2 and Table S3 in the Supplementary Materials with changes highlighted). We also present % of perforin, 4-1BB, IL-10 and IFN-g expressing cells. In addition, we show proportions of NK cell subsets and present expression of their effector proteins (perforin and granzyme A). We are convinced that this type of data presentation improved the paper and its clarity significantly. We also hope that now the uniqueness of the analytical approach is better visible.

At the end we would like to thank again the Reviewers for the suggestions and comments. We would like to thank the Reviewers for this joint work on the Manuscript and we hope the current version of the Manuscript will be endorsed by the Reviewer for the publication and accepted by the Editors.

Yours sincerely

Natalia Marek- Trzonkowska

Round 3

Reviewer 1 Report

The case study had been improved